# Response of Alfalfa (*Medicago sativa* L.) to Abrupt Chilling as Reflected by Changes in Freezing Tolerance and Soluble Sugars

**Hongyu Xu, Zongyong Tong, Feng He and Xianglin Li \***

Laboratory of Grassland Ecology and Management, Institute of Animal Sciences, Chinese Academy of Agricultural Sciences, No. 2 Yuanmingyuan West Road, Haidian district, Beijing 100193, China; xhycaas@163.com (H.X.); tongzongyong@caas.cn (Z.T.); hefeng@caas.cn (F.H.)

\* Correspondence: lxl@caas.cn; Tel.: +86-010-62815997

**Abstract:** Abrupt-chilling events threaten the survival of alfalfa plants, the ability to cope with such condition should be considered during cultivar selection in the production. To assess biochemical and molecular responses of alfalfa to abrupt chilling, the cultivars "WL440HQ" (WL) and "ZhaoDong" (ZD) were subjected to a five-phase experimental regime that included two abrupt-chilling events. The freezing tolerance of the crown was determined as the semi-lethal temperature ($LT_{50}$) calculated from electrolyte leakage. Soluble sugar concentrations were quantified by ion chromatography. The mRNA transcript levels of four genes encoding enzymes (β-amylase, sucrose phosphate synthase, galactinol synthase, and stachyose synthase) involved in sugar metabolism and two cold-regulated genes (*Cas15A* and *K3-dehydrin*) were quantified using quantitative real-time PCR analysis. During the abrupt-chilling events, the $LT_{50}$ decreased significantly in ZD but not in WL. The rapid response of ZD to abrupt chilling may have been due to the large increases in raffinose and stachyose concentrations, which were consistent with increased transcript levels of the galactinol synthase and stachyose synthase genes. Transcript levels of the cold-regulated genes *Cas15A* and *K3-dehydrin* were correlated with increased freezing tolerance under abrupt chilling. The results provide a reference for selection of appropriate cultivars to reduce the risk of crop damage in production areas where early autumn or late spring frosts are likely.

**Keywords:** alfalfa; fall dormancy; abrupt chilling; freezing tolerance; sugar metabolism; cold-regulated genes

---

## 1. Introduction

Alfalfa (*Medicago sativa* L.) is widely cultivated over more than 4 million ha in China. In the north, the main cultivation area, low temperatures occur in winter and frost occurs frequently in spring. Cold, a major factor limiting alfalfa production and distribution, causes serious economic losses. Two types of cold stress cause damage, namely, extremely low temperatures and abrupt chilling, of which the latter is exemplified by early autumn and late spring frosts [1,2]. The ability of alfalfa to resist low temperatures in winter is closely associated with fall dormancy [3], cold acclimation, and cold hardening [4]. Tolerance of alfalfa to gradually lowering temperatures is improved by cold acclimation [5]. When temperatures increase, the acquired cold resistance of alfalfa is lost through the process of deacclimation [6]. Nevertheless, the contents of cryoprotectant substances are much lower before cold acclimation (in early autumn) or after deacclimation (in late spring) [7–9]; at such stages, alfalfa is likely to be killed by frost [1]. Cold-resistant cultivars generally show low fall dormancy [10]. To increase winter survivability, cultivars differing in degree of fall dormancy have been selected

for production on the basis of tolerance to extremely low temperatures [9]. Although the ability to cope with abrupt-chilling events should be considered during cultivar selection, field studies for development of relevant assessment strategies are lacking.

The contents of many biochemical and metabolic process-related compounds change during cold acclimation and confer freezing tolerance on alfalfa [11]. Soluble sugars are among the most cold-sensitive compounds [12]. Soluble sugars act as osmoregulators [13], cryoprotectants [14], and signaling molecules [15] to stabilize the cell membrane and cell turgor [16] and scavenge reactive oxygen species [17]. The soluble sugar content of alfalfa crowns increases almost two- to three-fold after treatment with low temperature for only a few hours [18]. Sucrose accounts for as much as 80–90% of accumulated soluble sugars in alfalfa [18]. However, raffinose and stachyose are the most effective cryoprotectant sugars for protecting alfalfa [5,19,20]. Changes in soluble protein content are also closely associated with freezing tolerance of cold-acclimated alfalfa; for example, *K3-dehydrin*, a water-stress-protective protein, can maintain cell membrane integrity under cellular dehydration caused by cold stress [9,18,21].

In cold-acclimated alfalfa, soluble sugar accumulation is activated by specific enzymes [12]. Under cold conditions, the abundance of sucrose phosphate synthase (SPS), an enzyme involved in sucrose biosynthesis, varies among alfalfa cultivars that differ in cold resistance [18]. Galactinol synthetase (GaS) is a crucial enzyme in the synthesis of stachyose and raffinose [5,18,22]. The synthesis of these enzymes and other substances associated with alfalfa cold resistance are under direct or indirect genic control [23]. Increased expressions of cold-regulated genes, such as *Cas15A* [5,24] and *Cas18* [9,10], can be induced by low temperature. *Cas15A* encodes a protein of 14.5 kDa that plays an important role in cold resistance [24]. Transcription of *Cas18* is associated with specific proteins, such as dehydrins, antifreeze proteins, and vegetative storage proteins [9].

In the present study, two abrupt-chilling treatments were applied to alfalfa before acclimation and after deacclimation. We investigated changes in the freezing tolerance of two cultivars, the weakly fall-dormant "ZhaoDong" and highly fall-dormant "WL440HQ", in response to abrupt chilling and to compare biochemical and genetic responses between the cultivars. Such information will be useful for alfalfa selection in areas where early autumn or late spring frosts are likely.

## 2. Materials and Methods

### 2.1. Materials

Two alfalfa cultivars, commonly grown in northern China, "WL440HQ" (WL; fall dormancy score ~6) and "ZhaoDong" (ZD; fall dormancy score ~2), which represented the range in fall dormancy scores of cultivars grown in the region, were selected for the study. Seeds of WL were purchased from Zhengdao Ecological Technology Co. (Beijing, China), whereas seeds of ZD were provided by the Institute of Animal Husbandry (Heilongjiang, China). The seeds were pre-cultivated on sandy soil in May at the Experimental Station of the Chinese Academy of Agricultural Sciences (Hebei, China). After sowing, the soil moisture was carefully maintained by irrigation every second day, and weeds and pests were thoroughly suppressed.

After 1 month, sets of four uniform plants were transplanted into polyvinyl chloride (PVC) pipes of diameter 10 cm and height 15 cm. The pipes were filled with a 100:30:55 (*w/w/w*) mixture of soil, perlite, and vermiculite. The water-holding capacity of the mixture ((wet weight − dry weight)/dry weight) was 151%. Subsequent experiments, which were conducted in the laboratory, included five phases of a temperature-controlled treatment, each of which was replicated five times. A total of 200 plants were used: 2 cultivars × 5 phases × 5 biological replicates × 4 plants per pipe. After transplantation, alfalfa plants were grown for 2 weeks in a growth room under the following conditions: 24 °C/20 °C (day/night), a 14 h/10 h (light/dark) photoperiod, and photosynthetic photon flux density of 300–400 μmol m$^{-2}$ s$^{-1}$.

## 2.2. Experimental Treatment Process

After growth for 2 weeks, all PVC pipes were randomly arranged in an LRH-200-GD low-temperature light incubator (Taihong Medical Instruments, Guangdong, China) for temperature-controlled experiments. During this period, plants were watered daily, soil moisture was maintained at a water-holding capacity of 75%, and PVC pipe positions were changed randomly. The five temperature-controlled experimental phases are detailed in Figure 1.

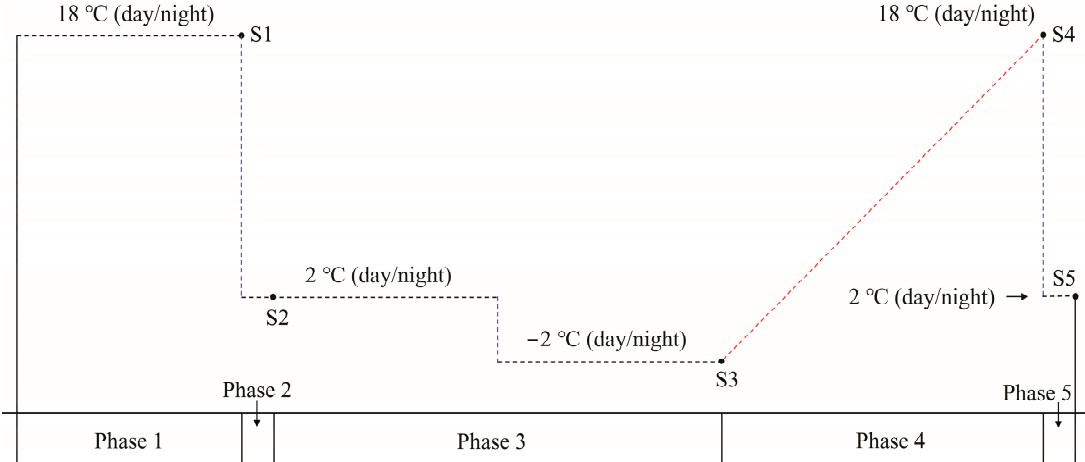

**Figure 1.** Phases 1 to 5 of the experimental treatment. Phase 1 consisted of 1 week of adaptive growth at 18 °C (day/night). During phase 2, the temperature was decreased to 2 °C (day/night) for one day. During phase 3, cold acclimation was carried out for 1 week at 2 °C (day/night), and the temperature was then decreased to −2 °C (day/night) for an additional week of cold hardening. Phase 4 involved increasing the temperature to 18 °C (day/night) at the rate of 2 °C/day to simulate deacclimation. In phase 5, the temperature was decreased to 2 °C (day/night) for one day. A 12 h photoperiod and a photosynthetic photon flux density of approximately 100–150 $\mu$mol m$^{-2}$ s$^{-1}$ were used during all five phases. S1 to S5 indicate sampling points at the end of each treatment phase.

## 2.3. Sampling

Five pipes (biological replicates) of each cultivar were randomly selected at the end of each phase shown in Figure 1, with pooled samples of crowns from four plants in each pipe used as one replicate. Sampling at each point was accomplished as early as possible during the day. Crowns from the four plants were sliced into small pieces (~2–3 mm) to obtain homogeneous samples for freezing tolerance tests and sugar and gene assessments. All sliced crowns from each pipe were divided into three segments. One segment was used for calculation of the semi-lethal temperature (LT$_{50}$) in a freezing test. The remaining two crown samples were immediately frozen in liquid nitrogen and stored at −80 °C for RNA extraction and sugar determination.

## 2.4. Freezing Tolerance

Freezing tolerance was evaluated based on the LT$_{50}$, the point at which the relative permeability of intracellular ions attained 50%, with measurements recorded as described by Anower and Sasaki [25,26]. The LT$_{50}$ values were determined for all five biological replicates of each cultivar. For the freezing test, fresh crown sections were placed into nine 2-mL centrifuge tubes. The tubes were stored at 4 °C for 2 h and then placed on ice for temporary preservation. The freezing test was conducted in a ZX-5C constant-temperature circulator (Zhixin, Shanghai, China) under a decreasing series of nine temperatures. To test the crown samples collected at the end of phases 1, 2, and 4, the temperature was decreased by −2 °C from −4 to −20 °C; for samples collected from phases 3 and 5, the temperature was decreased by −3 °C from −5 to −29 °C. Samples were maintained in an alcohol bath at each temperature for 1 h, and one frozen tube was removed after incubation at each temperature and thawed on ice

overnight. The next day, the tubes filled with crown samples were thawed at 4 °C for 2 h, and the crown slices were then transferred to 15 mL tubes, mixed with 5 mL deionized water, and shaken on a HZQ-A gyratory platform shaker (Hengrui Instrument and Equipment, Changzhou, Jiangsu, China) at 120 rpm for 12 h. The electrical conductivity ($EL_1$) of each sample was measured with a FE38 conductivity meter (Mettler Toledo, Shanghai, China). The samples were then autoclaved at 120 °C for 30 min, and electrical conductivity was remeasured ($EL_2$). Relative electrolyte leakage at a given freezing temperature and the $LT_{50}$ of a crown were calculated using formulas (1) and (2), respectively. In formula (1), EL is the electrical conductivity of deionized water; in formula (2), *x* represents the freezing temperature, *y* represents the relative electrolyte leakage, and A, B, and k are constants.

$$\text{Relative electrolyte leakage (\%)} = (EL_1 - EL)/(EL_2 - EL) \times 100 \qquad (1)$$

$$y\ (\%) = A/(1 + B \times e^{-kx}) \times 100 \qquad (2)$$

*2.5. Sugar Extraction and Determination*

Sugar concentrations were determined for all five biological replicates. A 0.1 g frozen crown sample was used for soluble sugars and starch determination. Soluble sugars were extracted by adding 0.7 mL of 80% ethyl alcohol solution and heating at 70 °C for 2 h, and the liquid supernatant was purified with chloroform. Soluble sugars were separated and analyzed on an Ion Chromatography System 5000 instrument (Thermo Fisher, Waltham, MA, USA) equipped with a Hypercarb PA20 column (Thermo Fisher), with a mobile phase consisting of $ddH_2O$ and 200 mM NaOH at a flow rate of 0.5 mL min$^{-1}$. The concentrations of soluble sugars were quantified based on external standards. We determined the concentrations of glucose, fructose, sucrose, raffinose, and stachyose and summed these values to obtain the total soluble sugars (TSS) content. The residue remaining after soluble sugar extraction was washed with methanol and measured for starch content through enzymatic hydrolysis [27].

*2.6. RNA Extraction and Quantitative Real-Time PCR*

Frozen crown samples were ground into powder in liquid nitrogen. Total RNA was extracted from the ground samples using the SG TriEx Extraction Kit (SinoGene Biotech, Beijing, China). Prior to cDNA synthesis, genomic DNA was removed with DNase I (Fermentas, Beijing, China), and PCR was performed to ensure that no genomic DNA remained. The cDNA was then synthesized from 1 µg of RNA using the First cDNA Synthesis kit (Thermo Fisher).

Quantitative real-time PCR assays were performed with a Step One PLUS system (Applied Biosystems, Foster City, CA, USA) and the SYBR Green Premix Kit (SinoGene). Relative gene expression levels were calculated using the $2^{-\Delta\Delta Ct}$ method [28]. The *Medicago sativa Actin* gene (JQ028730.1), which has been previously used as a reference gene for *M. falcata* and *M. truncatula* under cold or freezing conditions [29,30], was selected as a reference gene for normalization. Target and reference gene primers are listed in Table 1. In each set of five biological replicates, four replicates were randomly chosen to determine the relative expression level of related genes.

**Table 1.** Target gene (GOT) and reference gene (Ref) information. The GenBank accession numbers of *Medicago sativa* expressed sequence tags, used to design PCR primers, are also provided together with PCR primer sequences and amplified fragment sizes.

| Gen ID | Type | Sequence Homology | GenBank Accession NO | Primer Sequence (5'-3') | Size |
|--------|------|-------------------|----------------------|-------------------------|------|
| *SPS* | GOT | Sucrose phosphate synthase | AF322116.2 | CGCCTATTTGTGGGTGACTT TCGTTGCTCTCACCCTTCTT | 125 |
| *GaS* | GOT | Galactinol synthase | AY126615.1 | CTTGTTCTGGCCATGTTGTG TCCACACCTGTGTACCTCCA | 110 |
| *StaS* | GOT | Stachyose synthase | AY468361.1 | TGATCCAATGGGAGCTTTTT CCCAATCAGGTCGAATCATC | 95 |

**Table 1.** *Cont.*

| Gen ID | Type | Sequence Homology | GenBank Accession NO | Primer Sequence (5'-3') | Size |
|--------|------|-------------------|----------------------|-------------------------|------|
| *β-am* | GOT | β-amylase | AF026217.1 | TGGAGGAAATGTAGGGGATG TCCTAATACCGGAGCGATTG | 107 |
| *Cas15A* | GOT | Cas15A | L12461.1 | ATTTGCCGACAAGATCAAGG CCATGTTCATGACCCTCTCC | 102 |
| *K3-dehydrin* | GOT | K3-dehydrin | JX460852.1 | GGTGCTAGTGGTGCTGGT TGTCCTTGTCCATGTCCAGT | 113 |
| *Actin* | Ref | Actin | JQ028730.1 | TCGAGACCTTCAATGTGCCT ACTCACACCGTCACCAGAAT | 110 |

*2.7. Data Analysis*

Five biological replicates of both cultivars were sampled at each point (Figure 1). All five replicates were used in assessments of $LT_{50}$ and sugar concentrations, and four replicates were chosen randomly to determine the relative expression levels of genes. After determination, the $LT_{50}$, sugar content, and gene expression data were subjected to analysis of variance. In this study, two comparisons were made: (1) To evaluate the response of alfalfa to different environments during the five phases, comparison (from S1 to S5) was conducted for each cultivar; and (2) to compare the response of each cultivar to the same condition in the same phase (S1 vs. S1, S2 vs. S2, etc.). The analysis of variance was conducted using least significant difference test in SPSS software (SPSS, Chicago, IL, USA), with differences considered significant at $p < 0.05$. In addition, a Pearson correlation analysis between soluble sugar concentrations and $LT_{50}$ values was performed (Table 2) in SPSS, with significant differences assessed at $p < 0.05$ and 0.01.

**Table 2.** Correlation analysis between soluble sugar concentration and semi-lethal temperature ($LT_{50}$) in alfalfa. Significance differences at the 0.01 level are denoted by double asterisks.

| | | Total Soluble Sugar | | Sucrose | | Raffinose | | Stachyose | |
|---|---|---|---|---|---|---|---|---|---|
| | | WL | ZD | WL | ZD | WL | ZD | WL | ZD |
| $LT_{50}$ | Pearson correlation | −0.893 ** | −0.832 ** | −0.888 ** | −0.826 ** | −0.922 ** | −0.853 ** | −0.919 ** | −0.791 ** |

## 3. Results

*3.1. Freezing Tolerance*

The $LT_{50}$ of ZD, but not that of WL, decreased significantly in response to the two abrupt-chilling events (Figure 2). In phase 1, plants were cultured at room temperature, and no significant difference was observed between the two cultivars. Subsequently, the $LT_{50}$ of ZD decreased by 1.75 °C ($p < 0.05$) in response to abrupt chilling in phase 2 before cold acclimation, whereas WL exhibited no significant change ($p > 0.05$). During cold acclimation and hardening in phase 3 and during deacclimation in phase 4, the $LT_{50}$ of the two cultivars decreased and then increased significantly ($p < 0.05$). In phase 5, both WL and ZD recovered some of the freezing tolerance that was lost during deacclimation in phase 4. The freezing tolerance of ZD then significantly improved ($p < 0.05$) under abrupt chilling in phase 5, with the $LT_{50}$ decreasing by 9.85 °C; in contrast, the $LT_{50}$ of WL decreased non-significantly by 1.33 °C ($p > 0.05$). Together with its lower $LT_{50}$ and higher freezing tolerance, ZD responded more rapidly than WL to abrupt chilling to withstand cold stress. The $LT_{50}$ of ZD was significantly lower than that of WL at the end of phases 2 to 5 (Figure 2).

*3.2. Starch and Soluble Sugars*

3.2.1. Starch

Starch concentrations in WL and ZD in all five phases exhibited similar trends (Figure 3) and were consistent with changes in $LT_{50}$. Starch concentration decreased in phases 2 and 3, increased in phase 4,

and decreased again in phase 5. In WL, the starch concentration changed significantly ($p < 0.05$) only during deacclimation in phase 4. In ZD, significant changes ($p < 0.05$) were observed during abrupt chilling in phase 2, during cold acclimation and hardening in phase 3, as well as during deacclimation in phase 4. Under abrupt chilling after deacclimation, however, starch concentrations did not change significantly. At the start of the treatment, the starch contents of ZD were significantly higher than those of WL at S1 and S2 (Figure 3).

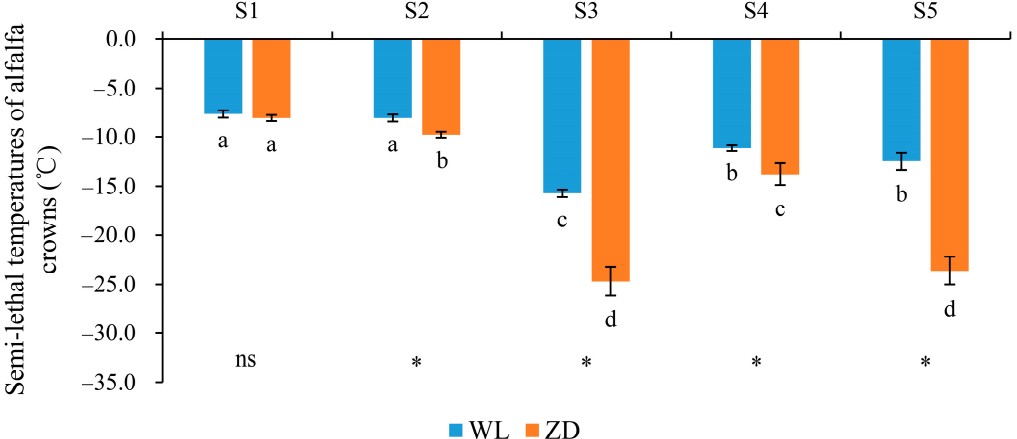

**Figure 2.** Semi-lethal temperature ($LT_{50}$) of alfalfa crowns at five sampling points. Mean values ($n = 5$) ± standard errors of the mean are shown. Different letters (a, b, etc.) below bars indicate a significant difference among the five sampling points (from S1 to S5) in one cultivar ($p < 0.05$). The asterisk indicates a significant difference between the cultivars (S1 vs. S1, S2 vs. S2, etc.) at one same sampling point ($p < 0.05$), and the 'ns' indicates no significant difference. S1 to S5 are the sampling points after the five phases of the temperature-controlled treatment regime: in phase 1 the temperature was maintained at 18 °C, in phase 2 it decreased to 2 °C, in phase 3 it was maintained at 2 °C and −2 °C for two weeks, in phase 4 it increased to 18 °C, and in phase 5 it decreased to 2 °C. WL = WL440HQ, ZD = ZhaoDong.

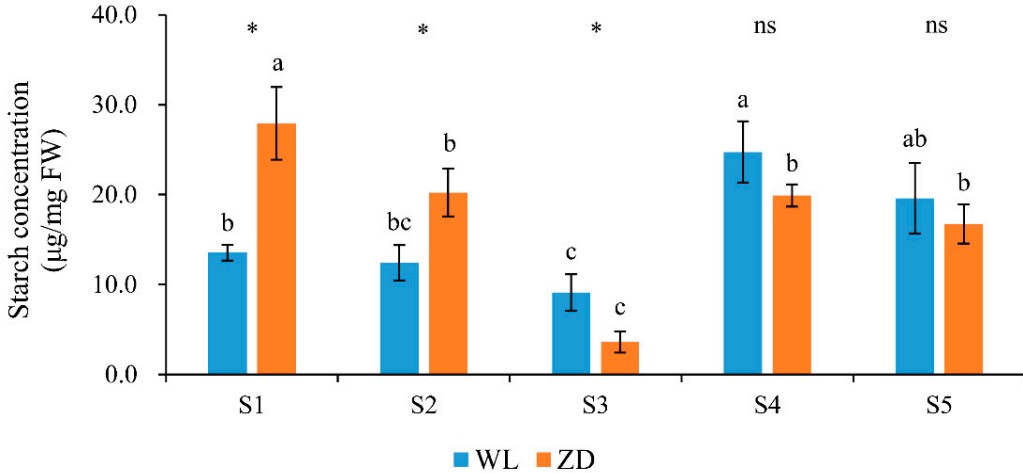

**Figure 3.** Starch concentrations in alfalfa crowns at five sampling points. Mean values ($n = 5$) ± standard errors of the mean are shown. Different letters (a, b, etc.) above bars indicate a significant difference among the five sampling points (from S1 to S5) in one cultivar ($p < 0.05$). The asterisk indicates a significant difference between the two cultivars (S1 vs. S1, S2 vs. S2, etc.) at one same sampling point ($p < 0.05$), and the 'ns' indicates no significant difference. S1 to S5 are the sampling points after the five phases of the temperature-controlled treatment regime: in phase 1 the temperature was maintained at 18 °C, in phase 2 it decreased to 2 °C, in phase 3 it was maintained at 2 °C and −2 °C for two weeks, in phase 4 it increased to 18 °C, and in phase 5 it decreased to 2 °C. WL = WL440HQ, ZD = ZhaoDong.

### 3.2.2. Soluble Sugars

Soluble sugars, especially raffinose and stachyose, showed marked responses to low-temperature treatments. The TSS, sucrose, raffinose, and stachyose contents exhibited similar trends at each phase; they increased in phases 2 and 3, decreased in phase 4, and increased in phase 5 (Figure 4), which was in contrast to the changes in $LT_{50}$. Correlation analysis revealed that the concentrations of TSS and the three soluble sugars (sucrose, raffinose, and stachyose) were all significantly negatively correlated ($p < 0.01$) with $LT_{50}$ (Table 2).

No significant difference in sucrose concentration was detected between the two cultivars at any sampling point. At the end of the five phases, sucrose accounted for the highest proportion of TSS (81.49–91.65% in WL and 89.05–95.87% in ZD). The concentration of TSS in ZD crowns was lower than that in WL crowns, and significant differences were observed at sampling points S1 to S3.

In both cultivars, the concentrations of raffinose and stachyose were low at sampling point S1. After abrupt chilling, cold acclimation, and deacclimation at points S2 to S5, raffinose and stachyose concentrations in ZD crowns were markedly higher ($p < 0.05$) than those in WL crowns. In ZD, raffinose and stachyose concentrations increased significantly in response to abrupt-chilling events. Raffinose and stachyose were the most sensitive soluble sugars in response to low temperature.

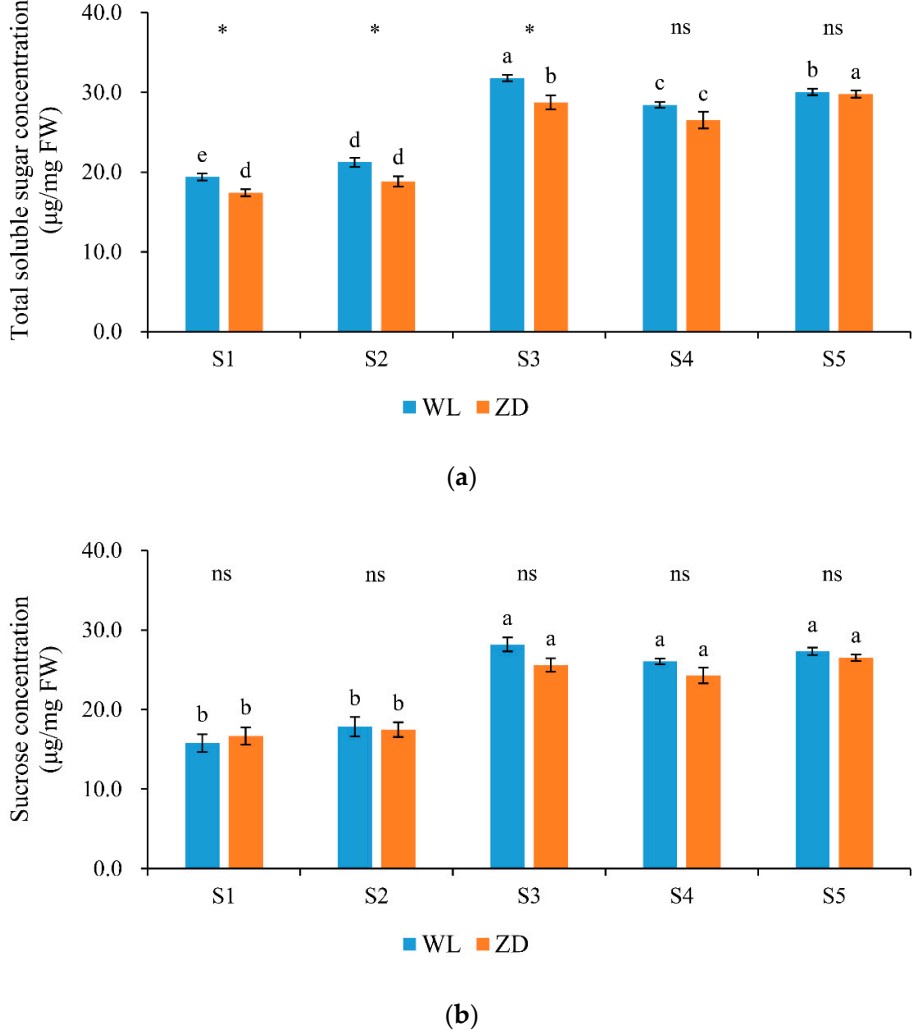

**Figure 4.** *Cont.*

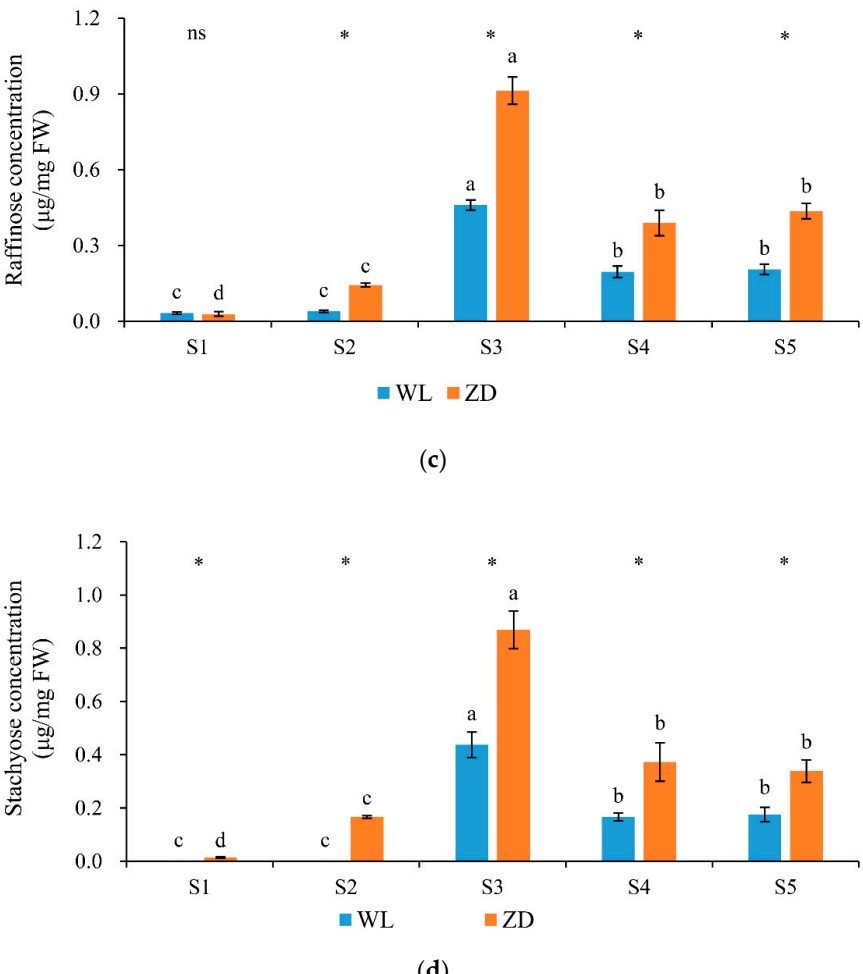

**Figure 4.** Soluble sugar concentrations in alfalfa crowns at five sampling points. Mean values (*n* = 5) ± standard errors of the mean are shown. Different letters (a, b, etc.) above bars indicate a significant difference among the five sampling points (from S1 to S5) in one cultivar (*p* < 0.05). The asterisk indicates a significant difference between the two cultivars (S1 vs. S1, S2 vs. S2, etc.) at one same sampling point (*p* < 0.05), and the 'ns' indicates no significant difference. S1 to S5 are the sampling points after the five phases of the temperature-controlled treatment regime: in phase 1 the temperature was maintained at 18 °C, in phase 2 it decreased to 2 °C, in phase 3 it was maintained at 2 °C and −2 °C for two weeks, in phase 4 it increased to 18 °C, and in phase 5 it decreased to 2 °C. (**a**) total soluble sugar; (**b**) sucrose; (**c**) raffinose; and (**d**) stachyose. WL = WL440HQ, ZD = ZhaoDong.

### 3.3. Gene Expression

#### 3.3.1. Sugar Metabolism-Related Genes

In the two cultivars, the relative expression level of *β-amylase* (*β-am*; Figure 5a) was low from S1 to S3. In ZD, significant changes were observed under abrupt chilling in phase 2 and during cold acclimation in phase 3. In both cultivars, *β-am* expression significantly increased as the temperature increased during deacclimation in phase 4. During abrupt chilling in phase 5, *β-am* expression decreased significantly in ZD.

The change in *SPS* expression (Figure 5b) in the two cultivars followed a similar trend to that of sucrose concentration. *SPS* expression in ZD and WL exhibited no obvious change in response to abrupt chilling in phase 2, but a significant increase was observed in response to abrupt chilling in phase 5.

*Galactinol synthase* (*Gas*) and *Stachyose synthase* (*StaS*) encode enzymes involved in raffinose and stachyose synthesis. In WL and ZD crowns, the expression of *GaS* (Figure 5c) was extremely low at S1 and S4, but increased significantly ($p < 0.05$) in response to abrupt chilling in phases 2 and 5 and almost attained a maximum. In both cultivars, no distinct change was observed during cold acclimation in phase 3, however, and *GaS* expression subsequently decreased to a minimum after deacclimation in phase 4. Of the four sugar metabolism-related genes analyzed, *GaS* was the most sensitive to abrupt chilling. The expression of *StaS* (Figure 5d) during the two abrupt-chilling treatments, in comparison, only increased significantly in ZD crowns during phase 2.

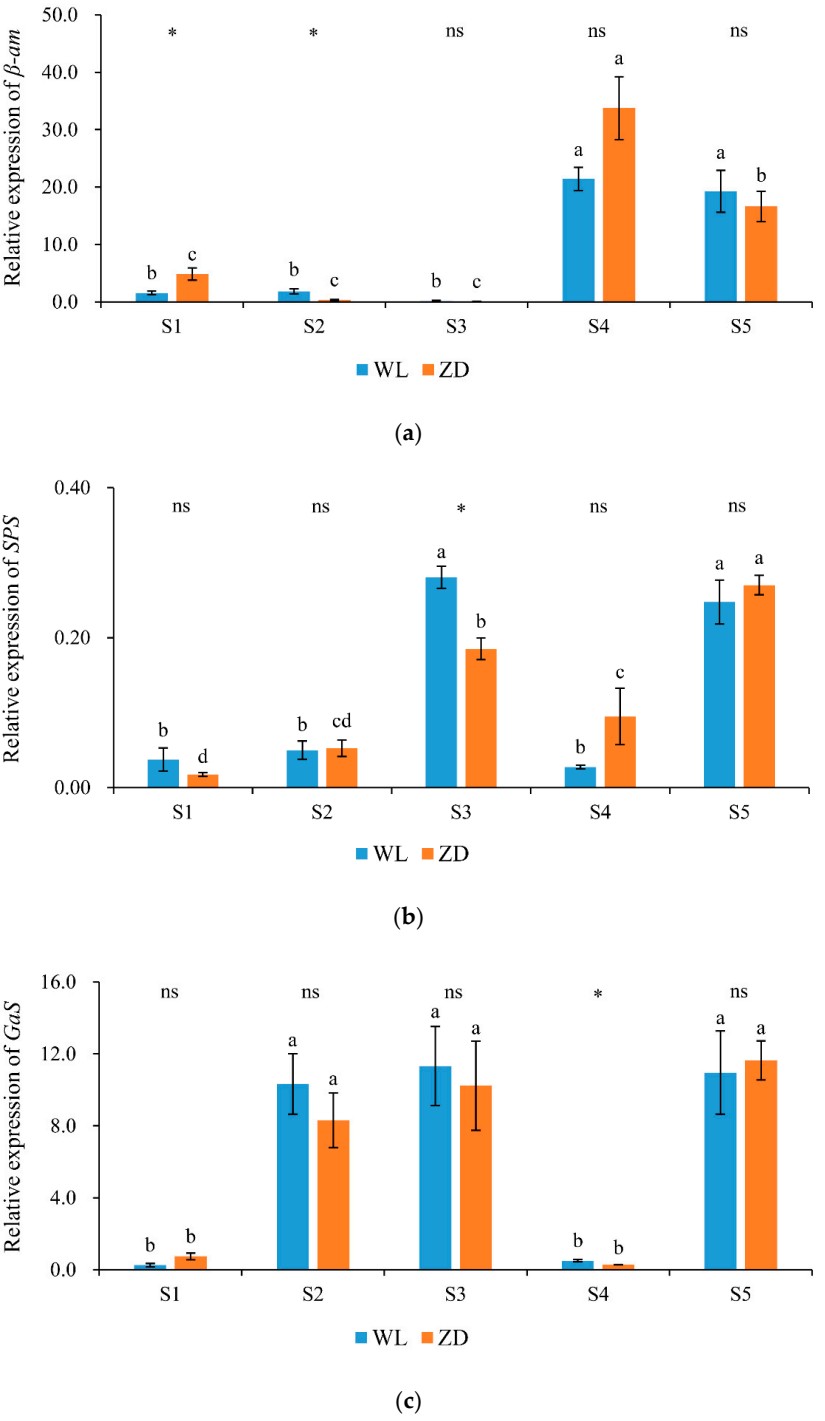

**Figure 5.** *Cont.*

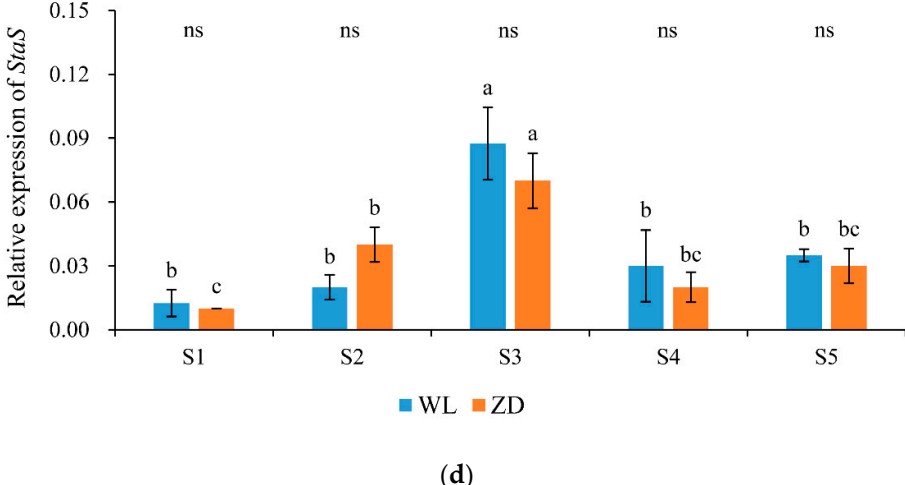

(**d**)

**Figure 5.** Relative expression levels of genes involved in carbohydrate metabolism at five sampling points. Mean values (*n* = 4) ± standard errors of the mean are shown. Different letters (a, b, etc.) above bars indicate a significant difference among the five sampling points (from S1 to S5) in one cultivar ($p < 0.05$). The asterisk indicates a significant difference between the cultivars (S1 vs. S1, S2 vs. S2, etc.) at one same sampling point ($p < 0.05$), and the 'ns' indicates no significant difference. S1 to S5 are the sampling points after the five phases of the temperature-controlled treatment regime: in phase 1 the temperature was maintained at 18 °C, in phase 2 it decreased to 2 °C, in phase 3 it was maintained at 2 °C and −2 °C for two weeks, in phase 4 it increased to 18 °C, and in phase 5 it decreased to 2 °C. (**a**) *β-am = β-amylase*; (**b**) *SPS* = Sucrose phosphate synthase; (**c**) *GaS* = Galactinol synthase; and (**d**) *StaS* = Stachyose synthase. WL = WL440HQ, ZD = ZhaoDong.

### 3.3.2. Cold-Regulated Genes

Throughout the five treatment phases, changes in *Cas15A* and *K3-dehydrin* expression levels (Figure 6) exhibited opposite trends to those of LT$_{50}$. In ZD and WL, relative expression levels of the two cold-regulated genes were significantly ($p < 0.05$) increased in response to the two abrupt-chilling treatments. In addition, *Cas15A* expression was significantly higher in ZD than that in WL at sampling points S1 and S4. The relative expression level of *K3-dehydrin* in WL was significantly higher than that in ZD at sampling points S3 and S5.

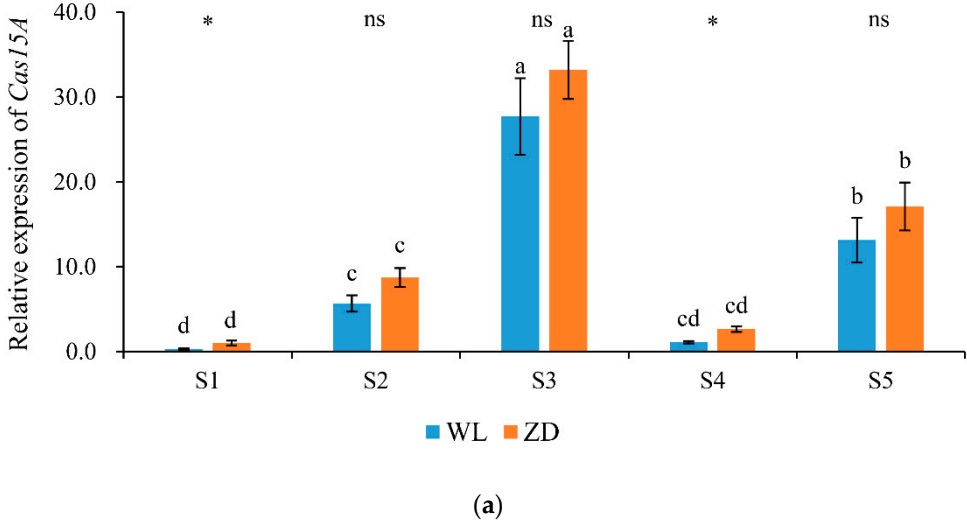

(**a**)

**Figure 6.** *Cont.*

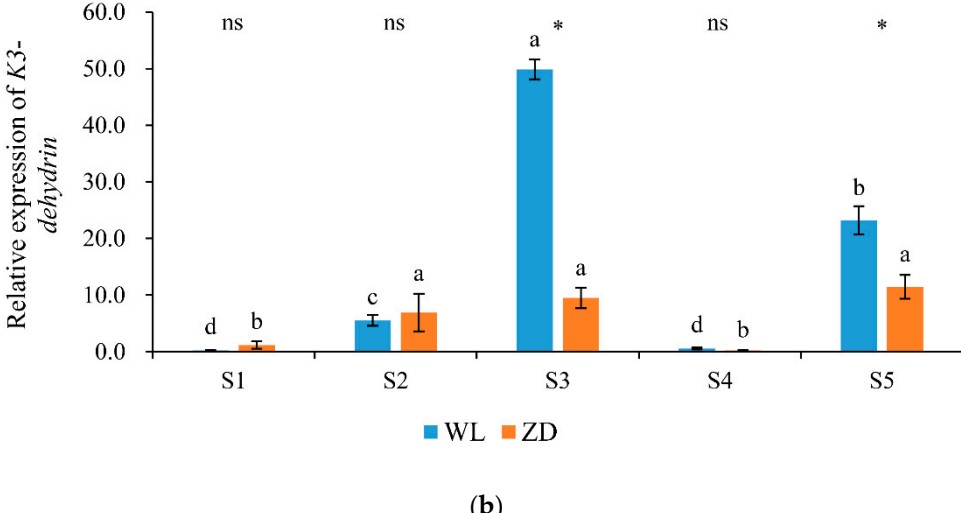

**(b)**

**Figure 6.** Relative expression levels of cold-regulated genes at five sampling points. Mean values (*n* = 4) ± standard errors of the mean are shown. Different letters (a, b, etc.) above bars indicate a significant difference among the five sampling points (from S1 to S5) in one cultivar (*p* < 0.05). The asterisk indicates a significant difference between the cultivars (S1 vs. S1, S2 vs. S2, etc.) at one same sampling point (*p* < 0.05), and the 'ns' indicates no significant difference. S1 to S5 are the sampling points after the five phases of the temperature-controlled treatment regime: in phase 1 the temperature was maintained at 18 °C, in phase 2 it decreased to 2 °C, in phase 3 it was maintained at 2 °C and −2 °C for two weeks, in phase 4 it increased to 18 °C, and in phase 5 it decreased to 2 °C. (**a**) *Cas15A*; (**b**) *K3-dehydrin*. WL = WL440HQ, ZD = ZhaoDong.

## 4. Discussion

In this study, the LT$_{50}$ of ZD decreased more rapidly than that of WL; ZD thus showed higher freezing tolerance to cope with the two abrupt-chilling events before cold acclimation and after deacclimation. The present results indicate that cultivars with low fall dormancy can resist the threat generated by temperature shocks, even in the absence of adequate cold acclimation, and are thus suitable candidates for selection breeding.

We analyzed soluble sugar concentrations in the crown, which is the critical overwintering organ [31], to evaluate the responses of alfalfa to cope with abrupt chilling. Mobilization of starch and accumulation of sucrose, raffinose, and stachyose suggest the importance of these substances in response to low temperature. Starch, which can be hydrolyzed to maltose by *β-am* and further converted to sucrose by a disproportionating enzyme [32,33], may be the source of the soluble sugars used to cope with cold stress. Furthermore, sucrose can be used to synthesize raffinose and stachyose [34]. In the present study, ZD showed a higher content of starch at the start of the treatment, and subsequently higher contents of raffinose and stachyose after acclimation and deacclimation. Similar changes in sugar composition have been reported in *Lolium temulentum* [35], *Arabidopsis* [36], and *M. sativa* [5,18] under cold stress. An extremely low expression level of *β-am* was observed before deacclimation in the current study, but a sharp increase in its expression and a significant improvement in starch content during deacclimation were detected. Whether starch hydrolysis is catalyzed by *β-am* cannot be determined from the present results, which is similar to a previous study of alfalfa in which no clear association was detected between *β-am* transcript abundance and starch hydrolysis [37].

Sucrose can stabilize freeze-dehydrated cells through direct contact with proteins and cell membranes [38]. *SPS* is a crucial enzyme involved in sucrose synthesis [18]. In the current study, the relative expression level of *SPS* and the sucrose concentration increased significantly during cold acclimation. In response to the two abrupt-chilling events, *SPS* expression only increased significantly after deacclimation, whereas the sucrose concentration did not change significantly during the same

period. Changes in sucrose synthesis were not completely consistent with changes in *SPS* expression, which may be associated with the fact that metabolite synthesis lags behind gene expression. On the basis of the present analysis, sucrose synthesis does not respond rapidly to abrupt-chilling events in alfalfa.

Previous studies have demonstrated that raffinose and stachyose accumulation in alfalfa is strongly associated with superior cold tolerance [18,19,22,39]. Throughout the present experiment, except for adaptive growth in phase 1, we observed markedly higher raffinose and stachyose concentrations together with notably higher freezing tolerance in ZD compared with those of WL (Figures 2 and 4). The present results support a previous observation [18] that differences in the maximum level of freezing tolerance between dormant (cold-resistant) and semi-dormant (cold-sensitive) cultivars are associated with raffinose and stachyose accumulation. In the previous analysis, ZD showed a higher starch content than WL at the start of the treatment, and subsequently higher contents of raffinose and stachyose after acclimation and deacclimation. Raffinose and stachyose may be synthesized and hydrolyzed from starch, thus the higher freezing tolerance of ZD in the present study might be partly associated with the higher starch content. During the two abrupt-chilling events in the present study, no large increase in raffinose and stachyose concentrations or freezing tolerance were observed in WL crowns. In comparison, the freezing tolerance of ZD increased significantly in response to both abrupt-chilling events, even though raffinose and stachyose concentrations in ZD crowns increased significantly only in response to abrupt chilling before cold acclimation. We speculate that this result may be due to the high concentrations of raffinose and stachyose remaining in ZD crowns after deacclimation at sampling point S4 (in phase 4 temperature increased to 18 °C). Taken together, the afore-mentioned results indicate that rapid accumulation of raffinose and stachyose in alfalfa can contribute to high freezing tolerance to cope with damage caused by abrupt chilling.

Galactinol synthase catalyzes the synthesis of galactinol, a galactose donor for raffinose and stachyose synthesis. Thus, GaS is a pivotal enzyme involved in the synthesis of raffinose and stachyose oligosaccharides [40]. In the two alfalfa cultivars, sharp increases in *GaS* expression were observed in response to the two abrupt-chilling events, but no significant change was detected during cold acclimation. Among the sugar metabolism-related genes analyzed in this study, *GaS* was the most sensitive in response to abrupt chilling. The changes in *GaS* expression are consistent with the conclusion that raffinose and stachyose are the soluble sugars most sensitive to abrupt chilling, and ZD showed higher concentrations of each sugar and superior freezing tolerance than WL. These results are congruous with a previous observation that *GaS* expression in alfalfa increased during cold acclimation and was maintained at a high level, which led to an accumulation of raffinose and stachyose, and in turn resulted in an increase in cold resistance [29]. Stachyose synthase is a vital enzyme in stachyose synthesis. During abrupt chilling in phase 2, *StaS* expression in ZD crowns was significantly increased. In response to the abrupt-chilling events, the relative expression levels of *GaS* and *StaS* in ZD were rapidly increased. These increases were accompanied by elevated synthesis of raffinose and stachyose, which may explain the rapid improvement in freezing tolerance of ZD.

In alfalfa, *Cas15A* is associated with cold resistance and serves as a molecular marker for cold-resistant cultivar selection [5]. In WL and ZD, *Cas15A* expression increased significantly in response to the two abrupt-chilling events. *K3-dehydrin* encodes a water-stress-protective protein that stabilizes cellular membranes against dehydration by promoting K-segment binding to proteins and cell membrane surfaces [21,41]. In the present study, *K3-dehydrin* expression increased significantly in both cultivars in response to the two abrupt-chilling events, and this change was accompanied by an increase in freezing tolerance. The expression of *K3-dehydrin* in WL was significantly higher than that in ZD at sampling points S3 (in phase 3 temperature was maintained at 2 °C and −2 °C for two weeks) and S5 (in phase 5 temperature decreased to 2 °C), which may be attributed to the lower freezing tolerance of WL. Lower freezing tolerance in the same cold environment implies the exertion of more severe water stress and thus stimulation of *K3-dehydrin* expression. The present results indicate that

the cold-regulated genes *Cas15A* and *K3-dehydrin* were strongly associated with increased freezing tolerance in response to the abrupt-chilling events.

## 5. Conclusions

Compared with the alfalfa cultivar with high fall dormancy, the cultivar with low fall dormancy showed a lower $LT_{50}$ and better freezing tolerance after abrupt chilling, acclimation, and deacclimation. The increased accumulation of raffinose and stachyose may play an important role in regulating cold-related characteristics, which was demonstrated by the increased expression levels of *GaS* and *StaS*. In addition, *Cas15A* and *K3-dehydrin* were involved in the adaptive responses to abrupt chilling in both cultivars. These results are beneficial for selection of appropriate cultivars to reduce the risk of damage caused by abrupt chilling in early autumn or late spring.

**Author Contributions:** Conceptualization X.L. and H.X.; methodology, Z.T.; software, F.H.; investigation, H.X. and X.L.; writing—original draft preparation, H.X. and X.L. All authors have read and agreed to the published version of the manuscript.

**Funding:** This research was funded by the National Key R&D Program of China (grant number 2016YFC0500608-2) and the Earmarked Fund for China Agriculture Research System (grant number CARS-34).

**Acknowledgments:** We thank Yuying Li and Yanjun Cui for their technical support during the experiment. We thank Liwen Bianji, Edanz Group China (www.liwenbiani.cn/ac), for editing the English text of a draft of this manuscript.

**Conflicts of Interest:** The authors declare no conflict of interest.

## Abbreviations

| WL44HQ | WL |
|--------|----|
| ZhaoDong | ZD |
| $LT_{50}$ | Semi-lethal temperature |
| TSS | Total soluble sugars |
| *β-am* | β-amylase |
| *SPS* | Sucrose phosphate synthase |
| *GaS* | Galactinol synthase |
| *StaS* | Stachyose synthase |

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
