# Peer review of "Response of Alfalfa (Medicago sativa L.) to Abrupt Chilling as Reflected by Changes in Freezing Tolerance and Soluble Sugars"

_agronomy, doi:10.3390/agronomy10020255_

Round 1
Reviewer 1 Report
The majority of my concerns are resolved. However, the authors’ response to my comment 1 “the whole experimental process was only performed once” still worries me a bit. Some other research findings with low-temperatures treatments can only be used as the rationale to design the current study, but not the solid evidence for supporting the conclusions in this study.
Author Response
Response to Reviewer 1 Comments
Point 1: The majority of my concerns are resolved. However, the authors’ response to my comment 1 “the whole experimental process was only performed once” still worries me a bit. Some other research findings with low-temperatures treatments can only be used as the rationale to design the current study, but not the solid evidence for supporting the conclusions in this study.
Response 1:
First of all, thank you again for your review. In another study, the cultivars WL440HQ and ZhaoDong were used. It was found that, after the treatment of adequate cold acclimation, ZhaoDong accumulated more soluble sugars (including raffinose and stachyose) and showed a larger freezing tolerance than that of WL440HQ.
In present study, WL440HQ and ZhaoDong were also used. During the treatment, cold acclimaiton, deacclimation and two abrupt chilling events were performed. To analysize the reasons of difference in freezing tolerance between the two alfalfa, the content of soluble sugars were compared. Results showed that, after the cold acclimation in phase 3, ZhaoDong owned higher content of soluble sugars (including raffinose and stachyose) accompanying by a larger freezing tolerance. These results are similar to the results of the previous study. In addition, five biological replicates were included as simultaneous trials in order to assess the consistency of the responses. In present study, the repeatability of the results is fine, the data is true and reliable as well. Thus, we hope these could dispel your concerns about the representativeness of our test results.
Reviewer 2 Report
My previous comments have all been sufficiently addressed. I thank the authors for their hard work on revising this manuscript. I now believe it is suitable for publication.
Author Response
Response to Reviewer 2 Comments
Point 1: My previous comments have all been sufficiently addressed. I thank the authors for their hard work on revising this manuscript. I now believe it is suitable for publication.
Response 1: Thank you again for your review, and best wishes for you.
This manuscript is a resubmission of an earlier submission. The following is a list of the peer review reports and author responses from that submission.
Round 1
Reviewer 1 Report
COMMENTS TO THE AUTHORS:
In this manuscript, the authors provide some evidence that the rapid response to abrupt chilling for the alfalfa cultivar WL440HQ may be due to the increase in raffinose and stachyose, which was further suggested by the enhanced expression of their related genes of Gas and StaS. This research is reasonably carried out and presented; however, I have a few suggestions for further improvement.
1: For experimental design, although 5 biological replicates were included in the study, it seems the whole experimental process has only been performed once. For answering any biological questions, it is crucial that the experiment needs to be independently carried out at least 2-3 times and similar results should be obtained.
2: On page 2, line 30, the overall rationale is not clear for the selection of two alfalfa cultivars (WL and ZD). Is the level 6 for WL is the highest? For an even more convincing conclusion, it will be interesting to select one more intermediate cultivar located between level 2 and level 6 and to see how it behaves.
3: Overall, the statistical analyses used in all the figures and comparisons are not clear. For example, is the comparison made among five different phases in one cultivar? Or among five phases between two different serovars? Please also explain the meaning of “a, b, c, d and e” labeled in the figures.
4: More improvement in language writing: for example: on page 1, line 25, what is the meaning for the “selection program”. On page 12, line 40-42, too strong conclusion of “……cold-related characteristics were due to increased accumulation of raffinose and stachyose ……”, it can be rephrased to: “……raffinose and stachyose may play an important role in regulating cold-related characteristics…....”
Reviewer 2 Report
This study that links together metabolism and cold response in two different cultivars of alfalfa. Right now the figures need to be redone as they are presented in a very confusing way. There may also be issues with the statistical test used, but it is hard for me to tell from the methods. I also have some minor concerns with the English used in the manuscript. Finally, while this manuscript adequately addresses the effect of temperature, it does not address the effect of cultivar in the written text but only in the figures.
Major comments:
1) Throughout the paper, the authors are making two comparisons: 1) S1-S5 in each cultivar and 2) S1 vs S1, S2 vs S2 etc between cultivars. While the ANOVA statistical test will capture all of these changes, the way these data are displayed in the figures right now is way too confusing for anyone to interpret. There are too many letters and additionally Figures 2-4 are done in a different style from 5. Throughout the paper, the figures need to be edited for clarity.
As an example take Figure 2. The "a" letter can be removed because it is a control. "bc" and "cd" should not exist as labels because there is no "c." Instead, the bars labeled "b", "bc", should be labeled with one symbol (say a dagger). The bars labeled "cd", "de" and "e" should be labeled with a different symbol (say a double dagger). Finally, the bars labeled "f" can be labeled with a *. Then you can present the figure by saying the bars with different symbols are significantly different from each other.
This process needs to be repeated for all the figures as there are multiple issues and errors with the way the letters are labeled.
2) I have a major issue with these sentences in section 2.7: "To evaluate the response of the two
alfalfa cultivars to different environments during the five phases and compare differences in their responses to the same condition at the same phase, a multiple range test was subsequently applied. The analysis of variance was performed through a least significant difference test in SPSS software." The multiple range test and least significant difference test are two different post hoc tests that will give different results. Which test was used? You cannot use both. You must choose one, and use that same test for all of your conclusions.
3) Why were 2 cultivars used? Why do they respond differently to the cold treatment? It would be good if the authors could elaborate on the differences between the two cultivars in the results and discussion. Right now, although the statistics are done comparing everything, the results only compare S1-S5 and not the effect of the cultivar.
3) Could the authors please elaborate on section 2.7 of the methods? I do not understand why 4 of the 5 replicates were chosen randomly for gene expression. Why not include the 5th replicate? Was there a technical issue?
4) Please ensure the standard error is being calculated correctly. Either your measurements are extremely precise, or there is an issue with the calculation because in some cases I cannot believe how small your error bars are with a sample size of only 4 or 5.
5) The labels S1-S5 are extremely uninformative. Figure 1 is nicely done, but it's really inconvenient for the reader to have to keep referring back to it. Instead, authors should give informative names to these points. One simple and easy fix would be to change the labels based on the temperature. (i.e. S1 = 18C, S2 = 2C, S3 = -2C, etc)
Minor comments:
1) There are many minor English issues throughout the manuscript. I highlight here the most glaring, but it would be good if the authors could get a native English speaker to thoroughly edit their manuscript. Most of the issues I see have to do with sentence structure, which tends to make the sentences long and confusing for readers to follow. This is most obvious in the Abstract, Introduction, and Discussion.
2) The title right now is a bit long, redundant, and confusing to me. It would be good if the authors could simplify the title.
3) The first sentence of the abstract is an incomplete sentence.
4) In the abstract, please abbreviate WL and ZD after first mentioning them to simplify the text.
5) Please restate the full names of WL and ZD when first mentioning them in the methods.
6) Please be consistent with figure legends. If you are going to abbreviate with WL and ZD in the text, please also do so in your figures.